# Screen time and chronic neck pain in Peru: A comparative population-based cross-sectional study in the COVID-19 post-pandemic period

Tomas Nakazato, Franco Romani-Romani ®*, César Gutiérrez

Facultad de Medicina Humana, Universidad de Piura, Lima, Peru

* franco.romani@udep.edu.pe

## Abstract

### Background

Chronic neck pain (CNP) is a prevalent condition worldwide. During the COVID-19 pandemic, its prevalence increased, partly due to the rise in screen time associated with digital device use. However, two years after the pandemic, there is still a lack of epidemiological data on its current prevalence. This study aimed to estimate the prevalence of CNP among adults in Peru in 2025 and assess its association with screen viewing time.

### Methods

We conducted a cross-sectional study involving a nationally representative sample of Peruvian individuals aged 18 years or older, whose data were collected in February 2025. The dependent variable —CNP— was self-reported based on symptoms experienced in the prior six months, and the exposure was the average daily screen time (hours/day). We assessed the association with multinomial logistic regression, adjusting by sex, age, rurality, and region of residence. The prevalences of the outcome and the exposure were described along with the estimates from November 2022, at the end of the pandemic.

### Results

1208 individuals were surveyed (51.6% women), with a mean age of 40.34 years (95% confidence interval [CI]: 39.3–41.3). CNP daily or near-daily affected 15.6% (95%CI: 13.4–18.0) of the participants; representing a slight but non-significant increase compared to 2022 (14.8%, 95%IC: 12.6–17.3) (p = 0.525). Screen time exceeding 8 hours per day was significantly associated with daily or near-daily CNP (odds ratio: 4.17, 95% CI: 2.15–7.69) compared with individuals who had never experienced neck pain.

**Data availability statement:** All relevant data are within the paper and its Supporting information files.

**Funding:** This research was funding by Universidad de Piura (proyect code: PI2505).

**Competing interests:** The authors have declared that no competing interests exist.

## Conclusions

Two years after the end of the COVID-19 social restrictions in Peru, the prevalence of CNP remains high, with 15 out of 100 adults experiencing this condition daily. The association between screen use and CNP was strongest in adults aged 18–29 years, while other factors may be more influential in adults ≥60. As prolonged screen use is a modifiable risk factor, reducing excessive exposure should be prioritized within prevention strategies to lower CNP prevalence.

## Introduction

The *Global Burden of Disease, Injury, and Risk Factor Study* (GBD) considers neck pain as pain that lasts at least one day, without specifying its intensity, duration or pattern of presentation over time [1]. Using this definition, since 1990, the prevalence of neck pain has increased by 77.3% worldwide, and by 2020, 203 million people reported being affected by this condition. However, between those years, the age-standardized prevalence has remained constant, with 2440 per 100,000 inhabitants reporting neck pain in 1990 and 2450 per 100,000 inhabitants in 2020 [2]. By 2021, the second year of the COVID-19 pandemic, of the 206 million people with neck pain, 42 million had newly developed the condition, and the age-standardized incidence was 519 per 100,000 inhabitants worldwide; however, in the Andes countries of Latin America, this incidence was lower (516 per 100,000 inhabitants) [3].

The International Association for the Study of Pain (IASP) defines chronic pain as that which is persistent or recurrent and that lasts longer than three months [4]. However, some studies have used six or more months to define chronic neck pain (CNP), which allows a more sensitive identification of individuals who experience pain for more than three months [5,6]. Physiologically, chronic pain persists beyond the normal time of resolution and, therefore, lacks the acute warning function of physiological nociception. From a clinical perspective, outcomes in patients with CNP are often suboptimal; among patients who experienced neck pain for six or more months, 53% did not achieve general symptom improvement after another six months of follow-up [7].

Regardless of the definition used, patients with CNP who do not receive treatment tend to develop persistently chronic disease. In a longitudinal study, patients with neck pain had a minimal reduction in pain intensity at six months of follow-up, whereas the difference was not substantial between six and 12 months [8]. Similarly, among patients with CNP who participated in clinical trials evaluating the effects of conservative treatments, those in the placebo or no intervention arm demonstrated a 10% reduction in the visual analogue pain scale scores at 10 weeks of follow-up, whereas at 52 weeks, no significant improvements were found [9]. These findings highlight the tendency of CNP to persist in the absence of proper diagnosis and intervention. Consequently, prevalence studies are essential for quantifying the burden of this condition and informing public health strategies.

Despite above mentioned, few studies have estimated the prevalence of CNP before and after the COVID-19 pandemic. A population study in Spain evaluated the presence of neck pain in the last 12 months and reported that, between 2006 and 2020, the lowest prevalence was recorded in 2020 (12.3%), whereas the previous measurement, made in 2017, was 14.6% [10]. In contrast, a population study in Peru reported an increase in the prevalence of CNP over time. Specifically, the percentage of people who reported daily neck pain in the prior six months increased from 6.1% —before the implementation of social restrictions due to the COVID-19 pandemic—to 14.8%—reported during the return to normal social interactions. Similarly, the percentage of individuals who reported experiencing CNP at least once a week increased from 12.4% to 27.8% [11]. In Peru, evidence is limited not only on changes in the prevalence of CNP, but also on the factors that may be contributing to the observed increase.

Some of these causes were consequences of the social restrictions imposed between March 2020 and 2022 by national governments to reduce community transmission of SARS-CoV-2. These restrictions led to significant shifts in people's lifestyles, including a marked increase in the time spent using electronic devices. A systematic review estimated that during the COVID-19 pandemic, 50.8% of adults increased their screen viewing time [12]. In China, 68.3% of people reported having increased the time spent viewing screens during the pandemic [13]. In Peru, one month after the end of pandemic-related social restrictions (November 2022), 12.3% of the population reported viewing screens for more than 8 hours per day [11]. Two years after the return to normalcy, however, we do not know if there have been changes in this practice among Peruvian individuals.

The impact of screen use on health is well documented in both children and adolescents [14], as well as adults [13]. This association became particularly evident during the COVID-19 pandemic. Between March 2020 and May 2023, the global increase in the use of electronic devices of various types was reported to have negatively affected the physical and mental health of multiple populations worldwide. One of the documented consequences was a rise in the prevalence of CNP [11,15,16], even several studies have reported an association between the prevalence of CNP and electronic device screen use [11,17,18]. However, since May 2023, when the World Health Organization declared the end of the pandemic phase of COVID-19, few observational studies have continued to estimate the prevalence of CNP. A study involving 302 adults from Singapore revealed that, between January and October 2023, 70.2% of participants reported experiencing nonspecific neck and shoulder pain. The study also identified that the time spent viewing cell phone screens was associated with the prevalence and intensity of neck pain [19].

Currently, no population-based studies conducted in the post-pandemic period have investigated changes in screen use patterns compared to the three years of social restrictions imposed during the COVID-19 pandemic. Consequently, it remains unknown whether such changes have influenced the prevalence of CNP at the population level. Given this context, our study aimed to estimate the prevalence of CNP in the adult population of Peru in February 2025, two years after the official end of the pandemic, and to assess its association with daily screen-viewing time. Additionally, we compared these findings with data collected in November 2022, immediately following the lifting of social restrictions, to explore potential temporal changes.

## Materials and methods

### Study design and scope

We conducted a survey-based cross-sectional study on February 20 and 21, 2025, in the Republic of Peru. This South American country is politically organized into 24 departments and the constitutional province of Callao. The total 2025-projected population is 34,350,244 inhabitants [20].

### Sample design

The target population consisted of the inhabitants of both the urban and rural areas of Peru aged 18 years and over. According to the number of individuals estimated to be able to vote in 2021, the population size was determined to be

24,290,921 people, of whom 79.9% reside in urban areas and 50.2% are women. The participants were selected according to the following eligibility criteria: (a) being an adult aged 18 years or older, (b) providing informed consent, and (c) being able to answer the survey questions.

The sample size was estimated with a margin of error of 2.8%, the maximum variance for population proportions (p = 0.50) and a confidence level of 95%, yielding a total of 1208 individuals. In the first stage, a stratified sample selection of clusters was performed, with the strata defined from the intersection between the geographic region (Lima, North Coast, South Coast, Central Coast, North Sierra, South Sierra, Central Sierra, and the East) and the scope (urban and rural). In the second stage, a sample of localities was randomly selected in each stratum, within which blocks were selected by means of systematic sampling with random start. In the third stage, systematic sampling with random start of houses were carried out in each selected block; finally, in the fourth stage, within each household, individuals were selected according to sex and age, until required distribution was achieved.

The sampling frame was based on the cartographic framework developed by the National Institute of Statistics and Informatics (Instituto Nacional de Estadística e Informática, INEI) during the 2017 national census. The sample ultimately covered 24 departments and the constitutional province and included participants from 101 districts located in 51 provinces.

**Instrument and variables**

The questions used to assess CNP and screen-viewing exposure were single-item measures and were not subjected to formal psychometric validation. However, both items were previously employed in epidemiological studies conducted in 2016 [21] and 2022 [11], providing indirect evidence of their suitability and practical utility for population level research.

The variable of interest was CNP, whose presence was determined by asking the participants the following question: *"Within the last six months, have you had pain in your neck, nape, and/or above your shoulders?"* The participants could respond with one of five responses: (1) no, I have never had neck pain or have had it very rarely; (2) yes, I have had neck pain once a month on average; (3) yes, I have had neck pain once a week on average; (4) yes, I have had neck pain daily or almost daily; and (5) not specified. The question was formulated based on the anatomical cervical region as defined by the Neck Pain Task Force [1]. A six-month recall period was used because pain persisting beyond three months exceeds the expected time for tissue healing or resolution of underlying conditions, thereby meeting the standard criteria for chronicity [4].

The exposure variable was the number of hours spent viewing electronic device screens, which was assessed by asking the following question: *"How many hours a day do you look at the screen of a device connected to the internet, such as a cell phone ("smartphone"), a laptop, a personal computer, a tablet and/or a television, for study, work or entertainment?"* The participants could respond to the question with one of the following categories: (1) I do not look at a screen or do so rarely, (2) I look at screens between 1 and < 4 hours/day, (3) I look at a screen between 4 and 8 hours/day, (4) I look at a screen > 8 hours/day, and (5) not specified. The cut-off points were selected by two main reasons. First, to facilitate direct comparison with a previous study conducted in the same population [11]. Second, because our aim was to assess whether prolonged screen viewing is associated with CNP, we defined exposure ≥8 hours per day as prolonged and therefore a potential risk factor. This threshold was informed by guidelines that limit the standard workweek to 48 hours [22] and by categories used in previous studies [23,24]. These considerations guided our analytic strategy, as epidemiological studies often employ varying cut-off points depending on the age group, the duration of exposure relative to the outcome, and the context in which screen use occurs.

The covariates were age in full years, which was categorized into 18 to <30 years, 30 to <60 years, and 60 years and over; sex; and area of residence (urban or rural). The macroregion of residence was categorized into Lima (which included Metropolitan Lima and the Constitutional Province of Callao), and the northern, central, southern, and eastern macroregions. Lima was considered a macroregion because 34.5% of the study population were from this region.

 

Socioeconomic level (SEL) was assessed with the methodology of the Peruvian Association of Market Intelligence Companies (Asociación Peruana de Empresas de Inteligencia de Mercado, APEIM) [25] from four dimensions: (a) variables related to the head of the household and the dwelling, including questions about the educational level reached, the health system with which they are affiliated, predominant material on the floors of the dwelling and access to a bathroom; (b) goods and services, including questions regarding ownership of a car or van for private use and paid domestic service; (c) household equipment and public services, including questions about ownership of a computer, refrigerator, and washing machine; and (e) public services, including questions about access to cable television and the internet. The scores of the questions ranged from 0 to 50, and the final categories were assigned an ordinal scale: SEL A (37–50 points), SEL B (22–36), SEL C (13–21), SEL D (8–12) and SEL E (≤ 7 points). SEL A represented individuals with the highest level of resources, while SEL E corresponded to those with the lowest.

## Survey administration

The survey was administered by IPSOS Opinion y Mercados SA through face–to–face interviews with the selected households. The respondents used the iField application (IPSOS Group SA) on a portable device to respond to the survey [26]; this application allows determination of the geolocation of the interviewers during field work, quality control of the data in real time, and assurance of adherence to the sample design described. The interviewers were residents with experience in the administration of such surveys and their instruments.

## Comparison of previous prevalences

The prevalences of CNP estimated in this study using the described methodology were compared with estimates for the same condition and target population that we published in a previous study [11]. We extracted the point prevalences and their corresponding 95% confidence interval (95% CI). Finally, we obtain three CNP estimates for three observation points that were depicted in a bar graph. The data for the first point, the most current one, were estimated with the methodology described above for February 2025. The data for the second and third points were obtained from our previous study based on a survey conducted on November 24 and 25, 2022 [11]. In that survey, the prevalence of CNP was similarly estimated prior to the start of social restrictions due to the COVID-19 pandemic with the following question: *"Did you experience any persistent pain in the neck, nape and/or above your shoulders in the six months prior to March 2020 (the start of COVID-19–related social restrictions)?"*

## Statistical analysis

After confirming the absence of reported missing data, we performed a descriptive analysis for all the sampling units. Age is reported as the mean and 95% confidence interval (95% CI). Categorical variables are reported as absolute counts and weighted proportions with their 95% CIs. We estimated the point prevalence of CNP using the pain frequency intensity categories described above. This ordinal scale allowed us to estimate the prevalence of experiencing (1) CNP once a month on average, (2) CNP once a week on average and (3) CNP daily or almost daily. This prevalence estimates used the total survey population as the denominator. In addition, we estimated these prevalences across strata defined by sex, age group, urban or rural residence, macro-region, socioeconomic status, and frequency of screen use. Weighted proportions yielded population-representative estimates for adults aged 18 years and older living in Peru in February 2025.

To evaluate the association between the independent variables and CNP, we used the chi-square test with second-order Rao–Scott correction. We evaluated the association of the exposure of interest (screen use) with CNP within the strata defined by sex and age group (18 to <30 years, 30 to <60, and ≥60 years). These stratification variables were selected to evaluate their role as potential confounders in the association of interest. In this analysis, we grouped the cumulative frequency of CNP into two categories, which allowed us to evaluate these confounders with respect to two forms of recurrent chronic pain, according to the definition of the IASP [27]. The CNP category "at least once a week"

included participants who responded: "daily pain" or "[pain] once a week on average"; and the CNP category "once a month" included those who responded "daily pain", "[pain] once a week on average" or "[pain] once a month on average". This analysis excluded participants who did not provide a response to the neck pain question.

To control for the confounding effects of the covariates, we formulated a multinomial logistic regression model, where the outcome was CNP categorized into four frequency levels. We defined the reference category as "No, I have never had neck pain or I have had it very rarely". The model included variables that showed statistically significant associations with CNP (p < 0.05) and the exposure in the crude analysis. Results were expressed as odds ratios (ORs) with their corresponding 95% confidence intervals (CIs). Nagelkerke's pseudo-R-squared was estimated to evaluate the goodness of fit of the model. We verified the assumption of independence of the observations and the absence of multicollinearity with the variance inflation factor (in which VIF > 10 indicated multicollinearity). Although the dependent variable is depicted with an ordinal scale, the ordinal logistic regression model did not meet the proportional odds assumption according to the Wald parallel lines test (F Wald = 3.418, p = 0.001), so we instead selected multinomial logistic regression [28].

Inferential analysis was performed with the survey data analysis module *svy* in STATA version 16. The described analyses were performed on a subpopulation of participants who provided valid answers to the questions (using the *subpop* command); that is, we excluded those who answered "not specified" to the questions that assessed CNP and screen time.

### Ethical aspects

The study protocol, including the use of verbal informed consent, was approved by the Institutional Research Ethics Committee of the University of Piura (file No. CIEI.0039.01). Trained interviewers obtained verbal informed consent from all participants after providing an explanation of the study objectives and before data collection. Participants indicated their willingness to participate by responding "yes" or "no," and these responses were recorded in the iField application on a portable electronic device. The survey was anonymous, and the statistical analysis was performed using an anonymized database. The survey did not include minors (<18 years old). This report was prepared in accordance with the STROBE guidelines for cross-sectional studies.

## Results

### Sample characteristics

A total of 1208 people with a mean age of 40.34 years (95% CI: 39.34–41.34) were interviewed. Of these, 15.2% were 60 years or older, and 51.6% were women. A total of 13.9% of the respondents reported that they do not look or rarely looked at electronic device screens, whereas 12.2% reported looking at screens for more than 8 hours/day. The average prevalence of CNP occurring once a month was 20.2%; while weekly CNP was reported by 14.7% of participants. The most severe form —daily occurrence— affected 15.6% of the sample (Table 1).

### Comparison of prevalence values

Compared with the estimated prevalence reported in November 2022, in February 2025, there was an increase of 0.81 percentage points in the prevalence of daily CNP and of 1.73 percentage points in the prevalence of CNP once a week on average. These differences were not statistically significant (p = 0.525). The prepandemic prevalence values were lower than those post pandemic: 6.1% for CNP daily, 6.3% for CNP once a week on average and 10.2% for CNP once a month on average (Fig 1A). For all patients who reported CNP at least once a month (daily + weekly + monthly), the prevalence among the participants in February 2025 was 51.2%, representing a non-significant increase of 1.6 percentage points over the November 2022 value (49.5%) (p = 0.504). Similarly, the prevalence of experiencing CNP at least once a week (daily + weekly) increased by 2.7 percentage points between the two time points (30.7% in 2025 versus 28.0% in 2022, p = 0.220) (Fig 1B).

**Table 1. General characteristics, screen device use and chronic neck pain in the Peruvian population.**

| Variable | n | Weighted percentage (%) | 95% Confidence interval | |
|---|---|---|---|---|
| | | | LL | UL |
| Age group (years) | | | | |
| 18 to <30 | 373 | 31.0 | 28.1 | 34.1 |
| 30 to <60 | 651 | 53.8 | 50.5 | 57.0 |
| ≥60 | 184 | 15.2 | 13.1 | 17.7 |
| Sex | | | | |
| Male | 593 | 48.4 | 45.2 | 51.7 |
| Female | 615 | 51.6 | 48.3 | 54.8 |
| Residential area type | | | | |
| Urban | 1002 | 79.7 | 76.3 | 82.7 |
| Rural | 206 | 20.3 | 17.3 | 23.7 |
| Macroregion | | | | |
| Lima | 500 | 34.0 | 31.3 | 36.9 |
| Northern | 268 | 23.9 | 21.1 | 26.8 |
| Central | 126 | 12.5 | 10.3 | 15.1 |
| Southern | 187 | 17.2 | 14.7 | 20.1 |
| Eastern | 127 | 12.4 | 10.3 | 14.8 |
| Socioeconomic level | | | | |
| A | 34 | 2.0 | 1.5 | 2.9 |
| B | 210 | 11.2 | 9.7 | 12.9 |
| C | 461 | 33.3 | 30.5 | 36.2 |
| D | 275 | 25.4 | 22.8 | 28.3 |
| E | 228 | 28.0 | 24.8 | 31.5 |
| Screen viewing | | | | |
| Rarely or never uses screens | 145 | 13.9 | 11.7 | 16.5 |
| 1 to <4 hours/day | 621 | 53.9 | 50.7 | 57.1 |
| 4–8 hours/day | 268 | 19.2 | 17.0 | 21.7 |
| >8 hours/day | 166 | 12.2 | 10.3 | 14.4 |
| Not specified | 8 | 0.8 | 0.4 | 1.6 |
| Chronic neck pain | | | | |
| No, I have never had neck pain or have had it rarely | 595 | 48.1 | 44.9 | 51.4 |
| Yes, I have had neck pain once a month on average | 243 | 20.2 | 17.7 | 23.0 |
| Yes, I have had neck pain once a week on average | 170 | 14.7 | 12.5 | 17.1 |
| Yes, I have had neck pain daily or almost daily | 184 | 15.6 | 13.4 | 18.0 |
| Not specified | 16 | 1.5 | 0.8 | 2.5 |

LL: lower limit, UL: upper limit.

(A) Frequency categories according to the answers to the survey question. (B) Cumulative frequency categories of chronic neck pain.

The category "at least once a week" includes participants who responded that they experienced neck pain daily or once a week on average; the category "once a month" includes participants who responded that they experienced pain daily, once a week on average or once a month on average. The p-values above the bars represent the comparison of prevalence between the two time points of interest using the Chi-square test with the second-order Rao–Scott correction.

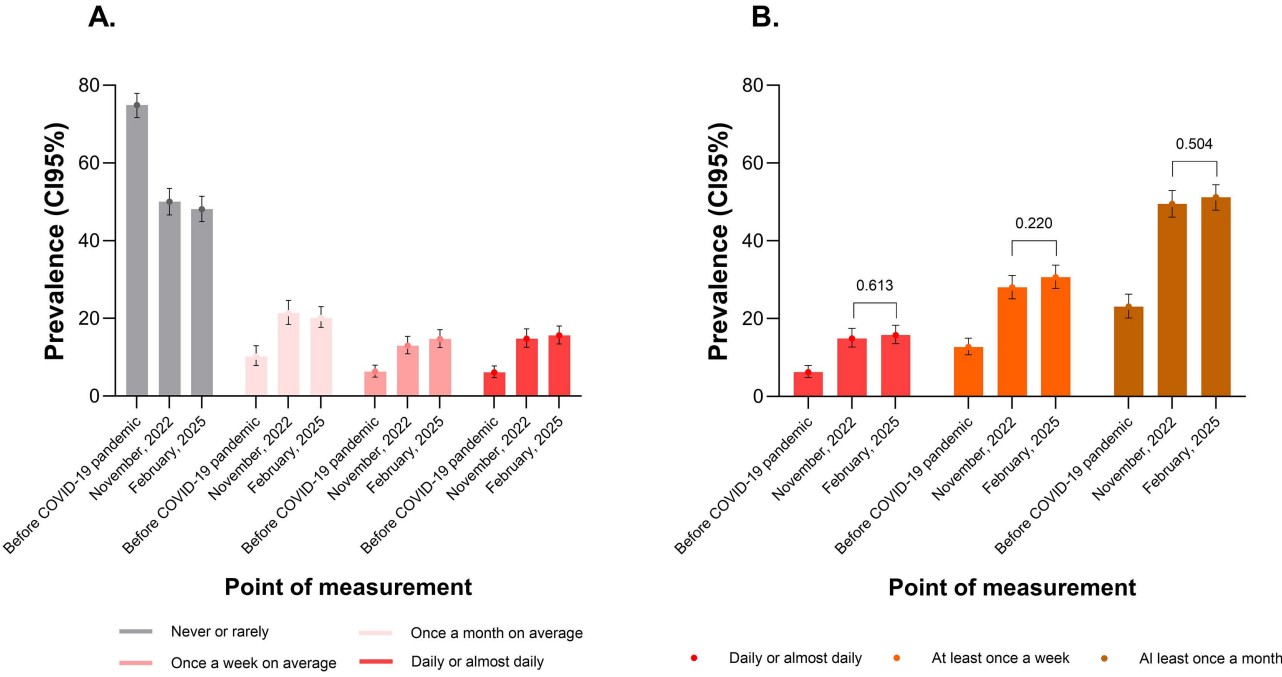

**Fig 1. Prevalence of chronic neck pain at three measurement points according to frequency in Peru.**

The proportion of individuals who reported not viewing screens decreased from 24.6% (95% CI: 21.7–28.8) in November 2022 to 14.0% (95% CI: 11.8–16.6) in February 2025. In contrast, the percentage of individuals who reported viewing screens for 1 to less than 4 hours per day increased from 41.4% (95% CI: 38.1–44.9) to 54.3% (95% CI: 51.1–57.5) (Fig 2).

## Crude analyses

The time spent viewing electronic devices screens was associated with the CNPs in the crude analyses. Among those who did not view screens, 58.4% did not report CNP. In contrast, among those who viewed screens for more than 8 hours per day, the prevalence of daily or almost daily CNP reached 20.2% (p = 0.002) (Fig 3A). Among people aged ≥60 years, 23.8% reported experiencing CNP daily or almost daily, whereas among those aged 18 to less than 30 years, the corresponding proportion was 9.3% (p = 0.010) (Fig 3B). In turn, age was associated with the number of hours spent viewing at electronic device screens. The proportion of people who did not view screens increased with age, from 5.7% among those aged 18 to < 30 years to 27.9% among those aged ≥60; in contrast, the proportion of those who viewed screens > 8 hours/day declined across older age groups (p < 0.001) (Fig 3C).

Among women, the prevalence of CNP once a week on average (16.7%) and daily (19.2%) was significantly higher than that among men (13.1% and 11.8%, respectively; p = 0.002). We also found that the macroregion of residence was associated with CNP; daily CNP affected 30% of the residents in the eastern region; in contrast, the largest proportion of respondents who reported never having experienced neck pain (56.8%) were residents of the southern region. Neither rurality nor socioeconomic level was associated with CNP (Table 2).

In the stratified analysis, we observed no significant increase in the prevalence of daily CNP with greater screen exposure in either sex. However, when stratifying by age group, an increasing prevalence of daily CNP with higher levels of screen viewing was observed among individuals aged 18 to under 30 years. This trend was not observed in the other

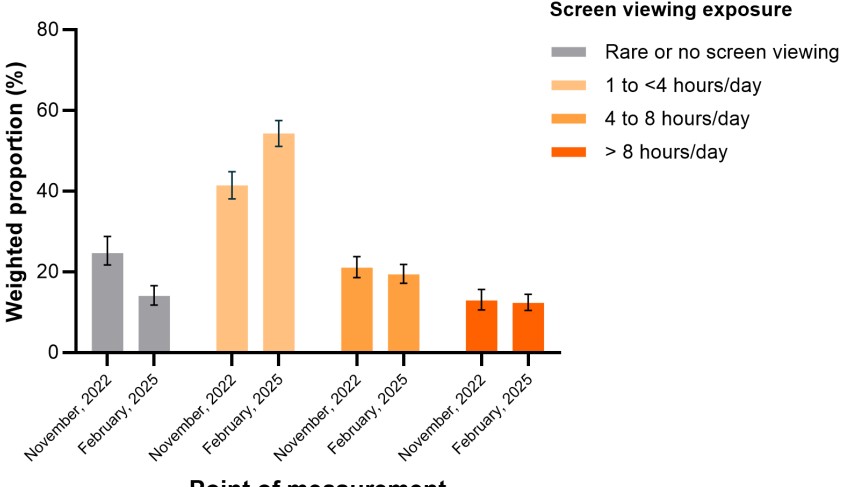

**Fig 2. Comparison of the weighted prevalence of electronic screen viewing exposure between November 2022 and February 2025 in the population of Peru.**

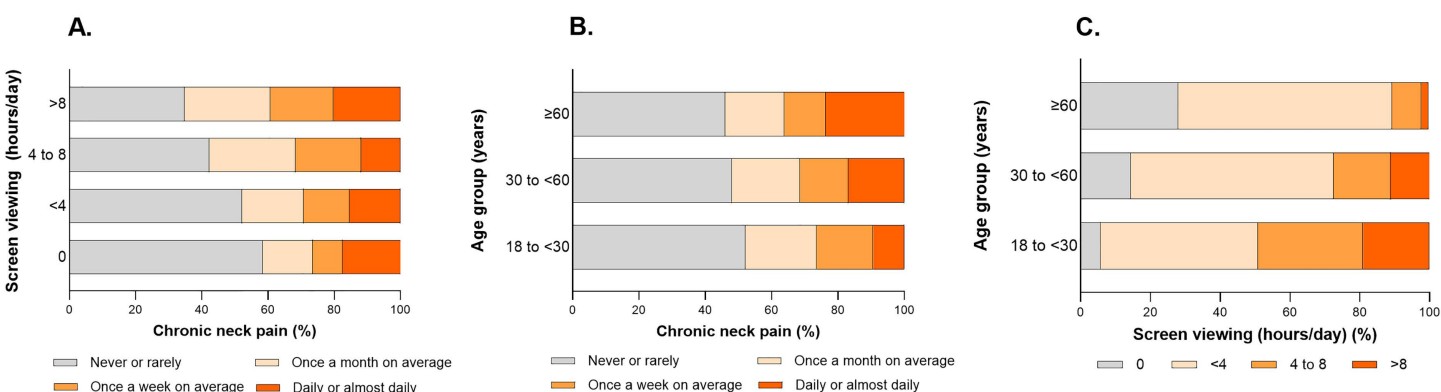

**Fig 3. Bivariate analysis of the association between screen viewing and chronic neck pain, considering age as a confounding variable.** (A) Proportion of chronic neck pain (outcome) across (exposure). (B) Proportion of chronic neck pain by age group (potential confounder). (C) Proportion of screen viewing (exposure) by age group (potential confounder).

age groups. Among women and individuals aged 18 to less than 30 years was observed a greater prevalence of at least weekly CNP with increasing screen time. Finally, the prevalence of at least monthly CNP increased with greater screen time among both men and women, among those aged 18 to less than 30 years and among those aged 30 to less than 60 years. Among individuals aged ≥60 years, no statistically significant association was found between screen time and the prevalence of CNP at any frequency (Table 3).

## Multinomial logistic regression

These findings offer strong evidence supporting the association between electronic screen viewing and CNP. Compared with not viewing screens, viewing screens 4–8 hours/day was associated with greater odds of experiencing CNP once a week and once a month on average. Additionally, viewing screens > 8 hours/day, compared with not viewing screens, was

**Table 2. Intensity of chronic neck pain according to sociodemographic strata and electronic screen viewing time.**

| Variable | N | Chronic neck pain | | | | p value |
|---|---|---|---|---|---|---|
| | | No, I have never had neck pain or have had it rarely | Yes, I have had neck pain once a month on average | Yes, I have had neck pain once a week on average | Yes, I have had neck pain daily or almost daily | |
| | | Weighted % (95%CI) | Weighted % (95%CI) | Weighted % (95%CI) | Weighted % (95%CI) | |
| Sex | | | | | | |
| Male | 585 | 54.5 (49.8, 59.2) | 20.6 (16.9, 24.8) | 13.1 (10.2, 16.7) | 11.8 (9.2, 15.1) | 0.002 |
| Female | 601 | 43.7 (39.2, 48.2) | 20.5 (17.0, 24.4) | 16.7 (13.6, 20.4) | 19.2 (15.8, 23.0) | |
| Age (years) | | | | | | |
| 18 to < 30 | 368 | 52.0 (46.1, 57.9) | 21.6 (17.1, 26.9) | 17.1 (6.6, 13.1) | 9.3 (6.6, 13.1) | 0.010 |
| 30 to < 60 | 638 | 48.0 (43.6, 52.4) | 20.6 (17.1, 24.6) | 14.5 (11.6, 18.0) | 16.9 (13.9, 20.5) | |
| 60 and over | 180 | 46.0 (38.0, 54.1) | 17.9 (12.5, 25.0) | 12.4 (7.9, 18.9) | 23.8 (17.5, 31.5) | |
| Residential area type | | | | | | |
| Urban | 986 | 49.3 (46.0, 52.6) | 19.2 (16.8, 21.9) | 15.0 (12.8, 17.5) | 16.5 (14.1, 19.2) | 0.371 |
| Rural | 200 | 47.5 (38.1, 57.1) | 25.6 (17.9, 35.2) | 14.9 (9.2, 23.4) | 12.0 (7.3, 19.2) | |
| Macroregion | | | | | | |
| Lima | 498 | 49.4 (45.0, 53.8) | 23.5 (19.9, 27.4) | 14.4 (11.6, 17.8) | 12.7 (10.1, 15.9) | 0.007 |
| Northern | 264 | 44.1 (37.4, 51.0) | 21.5 (16.1, 28.1) | 18.3 (13.4, 24.3) | 16.2 (11.7, 22.0) | |
| Central | 120 | 47.0 (36.6, 57.6) | 19.9 (12.7, 29.6) | 17.8 (11.0, 27.6) | 15.3 (9.1, 24.6) | |
| Southern | 183 | 56.8 (47.7, 65.5) | 19.0 (12.5, 28.0) | 13.3 (8.2, 20.8) | 10.9 (6.8, 16.8) | |
| Eastern | 121 | 48.0 (38.2, 57.9) | 12.6 (7.3, 20.8) | 9.5 (5.2, 16.8) | 30.0 (21.7, 39.7) | |
| Socioeconomic level | | | | | | |
| A | 34 | 70.8 (53.6, 83.6) | 20.4 (10.1, 37.2) | 8.8 (2.9, 23.9) | 0.0 | 0.202 |
| B | 209 | 48.2 (41.2, 55.2) | 22.8 (17.4, 29.3) | 15.1 (10.8, 20.8) | 13.9 (9.7, 19.6) | |
| C | 453 | 46.2 (41.2, 51.3) | 21.0 (17.2, 25.4) | 17.1 (13.5, 21.4) | 15.7 (12.3, 19.9) | |
| D | 271 | 52.1 (45.9, 58.2) | 15.3 (11.4, 20.2) | 15.4 (11.4, 20.4) | 17.3 (13.0, 22.6) | |
| E | 219 | 48.0 (40.4, 55.7) | 23.8 (17.7, 31.3) | 12.5 (8.1, 18.8) | 15.7 (11.2, 21.7) | |
| Screen viewing | | | | | | |
| Rarely or never uses screens | 142 | 58.4 (48.7, 67.4) | 15.1 (9.2, 23.9) | 9.0 (4.9, 16.2) | 17.5 (11.4, 25.8) | 0.002 |
| 1 to <4 hours/day | 615 | 52.1 (47.6, 56.7) | 18.6 (15.3, 22.5) | 13.8 (10.9, 17.4) | 15.4 (12.5, 18.9) | |
| 4–8 hours/day | 267 | 42.3 (36.0, 48.8) | 26.1 (20.8, 32.2) | 19.8 (14.9, 25.8) | 11.8 (8.2, 16.8) | |
| >8 hours/day | 162 | 34.8 (27.3, 43.2) | 25.9 (18.4, 35.2) | 19.1 (13.3, 26.5) | 20.2 (14.0, 28.2) | |

Estimated using the chi-square test with second-order Rao–Scott correction.

a significant risk factor for CNP at all frequencies. Furthermore, a gradient was observed in the odds ratios for CNP associated with screen viewing exceeding 8 hours per day: OR = 3.02 for CNP occurring once a month on average, OR = 3.87 for weekly CNP, and OR = 4.17 for daily or almost daily CNP (Fig 4).

Compared with individuals aged 18 to <30 years, no significant differences were observed in the odds of experiencing monthly or weekly CNP across the other age groups. However, for daily CNP, participants aged 30 to <60 years had an OR of 2.23, and those aged 60 years or older had an OR of 4.06. Compared with men, women showed higher odds of experiencing CNP on a weekly and daily basis, but not monthly. Additionally, residents of the eastern macroregion had

**Table 3. Stratified analysis of the associations between electronic screen viewing time and chronic neck pain by sex and age group.**

| Stratum/Exposure | | Neck pain daily or almost daily | | | Neck pain at least once a week | | | Neck pain at least once a month | | |
|---|---|---|---|---|---|---|---|---|---|---|
| | N | % | 95%CI | p value | % | 95%CI | p value | % | 95% CI | p value |
| **Sex** | | | | | | | | | | |
| *Male* | | | | | | | | | | |
| Rarely or never uses screens | 63 | 19.8 | 10.5, 34.3 | 0.849 | 29.1 | 17.8, 43.7 | 0.605 | 38.7 | 25.5, 53.9 | 0.017 |
| 1 to <4 hours/day | 307 | 10 | 7.0, 14.2 | | 22.9 | 17.8, 28.9 | | 42.9 | 36.6, 49.5 | |
| 4–8 hours/day | 137 | 10.4 | 5.8, 17.8 | | 25.4 | 18.3, 34.2 | | 48.2 | 39.1, 57.5 | |
| >8 hours/day | 78 | 15.6 | 8.2, 27.9 | | 30.2 | 19.9, 43 | | 59.4 | 46.7, 71.0 | |
| *Female* | | | | | | | | | | |
| Rarely or never uses screens | 79 | 16 | 9.0, 27.0 | 0.636 | 24.9 | 15.7, 37.2 | 0.009 | 43.5 | 31.7, 56.1 | <0.001 |
| 1 to <4 hours/day | 308 | 20.9 | 16.2, 26.6 | | 35.7 | 29.9, 42.1 | | 52.9 | 46.5, 59.2 | |
| 4–8 hours/day | 130 | 13.4 | 8.3, 20.9 | | 38.4 | 29.7, 48.0 | | 68.2 | 59.1, 76.1 | |
| >8 hours/day | 84 | 24.2 | 15.5, 35.6 | | 47.1 | 35.7, 58.9 | | 70.2 | 58.9, 79.5 | |
| **Age group** | | | | | | | | | | |
| *18 to <30* | | | | | | | | | | |
| Rarely or never uses screens | 17 | 0 | --- | 0.019 | 13.9 | 2.0, 55.6 | 0.008 | 13.9 | 2.0, 55.8 | <0.001 |
| 1 to <4 hours/day | 144 | 7.8 | 4.4, 13.6 | | 21.3 | 14.7, 29.9 | | 41.6 | 32.6, 51.3 | |
| 4–8 hours/day | 125 | 7.6 | 3.9, 14.3 | | 27 | 19.4, 36.2 | | 50.5 | 41.2, 59.8 | |
| >8 hours/day | 82 | 18.4 | 10.3, 30.9 | | 41.3 | 29.7, 53.9 | | 69.3 | 57.5, 78.9 | |
| *30 to <60* | | | | | | | | | | |
| Rarely or never uses screens | 77 | 18.1 | 10.3, 29.8 | 0.714 | 23.3 | 14.5, 35.2 | 0.055 | 40.2 | 28.4, 53.3 | 0.005 |
| 1 to <4 hours/day | 361 | 16.4 | 12.4, 21.3 | | 31.1 | 25.8, 37.0 | | 50.7 | 44.7, 56.6 | |
| 4–8 hours/day | 124 | 14.4 | 8.7, 23 | | 35.1 | 26.2, 45.1 | | 60.7 | 51.1, 69.5 | |
| >8 hours/day | 76 | 21.9 | 13.4, 33.6 | | 38.1 | 27, 50.7 | | 61.6 | 49.2, 72.6 | |
| *≥60* | | | | | | | | | | |
| Rarely or never uses screens | 48 | 23.9 | 12.5, 40.8 | 0.984 | 37.9 | 23.7, 54.5 | 0.809 | 56.2 | 40.0, 71.1 | 0.314 |
| 1 to <4 hours/day | 110 | 23.6 | 15.9, 33.5 | | 35.1 | 25.9, 45.5 | | 47.9 | 37.8, 58.2 | |
| 4–8 hours/day | 18 | 25.1 | 10.0, 50.2 | | 41.9 | 19.8, 67.8 | | 91 | 69.6, 97.8 | |
| >8 hours/day | 4 | 20.5 | 2.6, 71.6 | | 20.5 | 2.6, 71.6 | | 55.6 | 14.3, 90.4 | |

P value obtained with the adjusted Wald test in the binary logistic regression, taking electronic device screen use as the exposure variable.

Neck pain at least once a week: includes the response categories "pain once a week on average" and "daily or almost daily pain".

Neck pain at least once a month: includes the response categories "pain once a month on average", "pain once a week on average", " daily or almost daily pain".

higher odds of daily CNP (OR = 2.96, p<0.001) compared with those living in Metropolitan Lima. Overall, the model accounted for 10.4% of the variability in CNP frequency (Table 4).

## Discussion

In Peru, in February 2025, 15.6% and 14.7% of the Peruvian adult population reported experiencing CNP daily and once a week, respectively. These prevalences are slightly greater than those reported at the end of social restrictions imposed due to the COVID-19 pandemic [11]. Concurrently, the proportion of individuals who reported not viewing electronic device screens decreased from 24.6% in November 2022 to 14.0% in 2025. Screen viewing levels were clearly associated with CNP and its presentation pattern. The highest prevalence was observed among individuals who reported more than 8

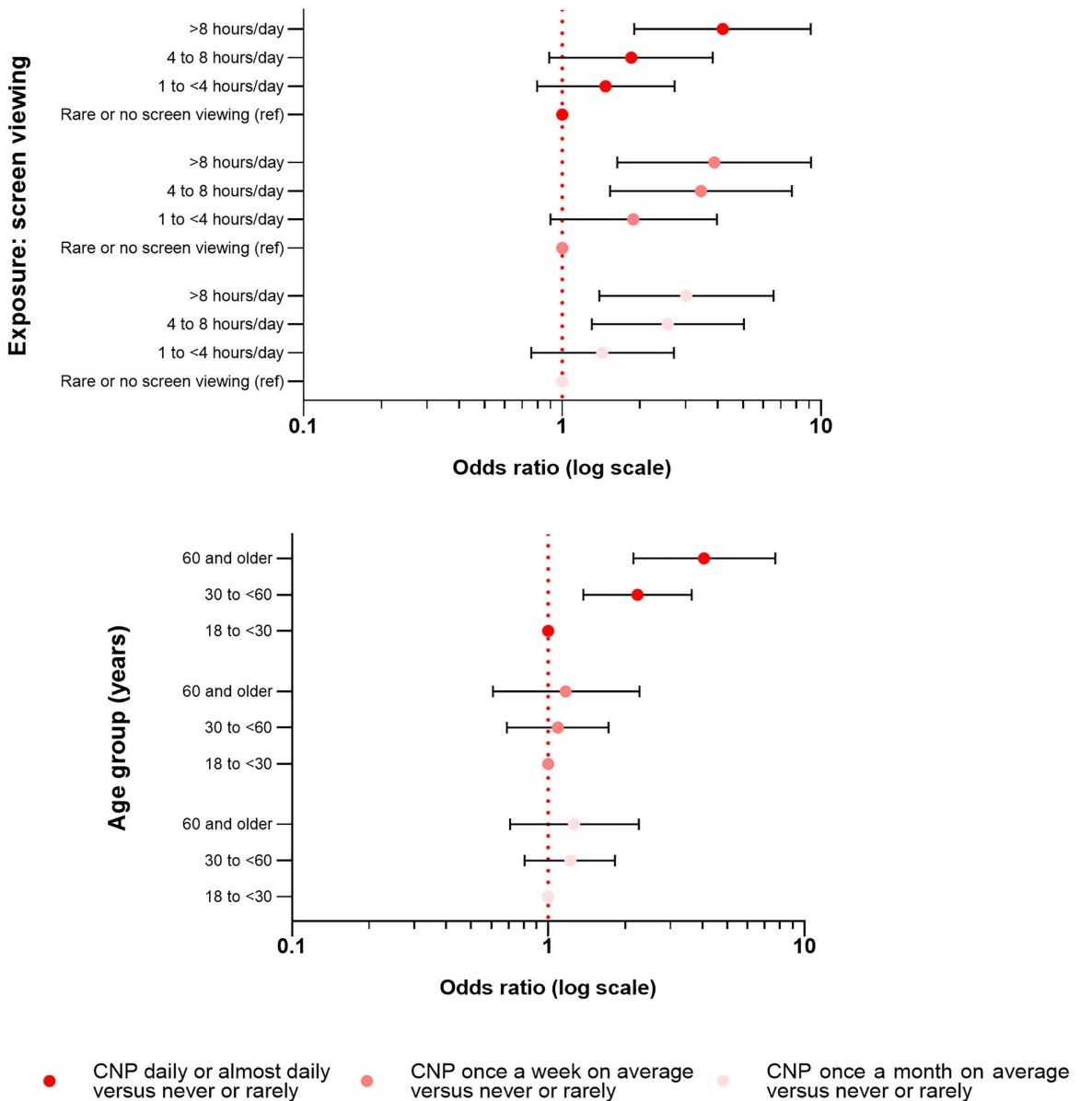

**Fig 4. Forest plot showing odds ratios for chronic neck pain according to electronic screen viewing exposure and age group.**

hours of screen use per day. This level of exposure was strongly associated with CNP across all intensity levels, regardless of sex or age group.

In this study, we estimated the prevalence of CNP for two years after October 2022, the date at which the social restrictions in Peru imposed due to the COVID-19 pandemic ended. These estimates provide us with a first approximation of the trend in experiencing CNP among the Peruvian population. In general, compared with November 2022, we observed a non-significant increase of 1.6 percentage points in the prevalence of CNP at least once a month, that is, the proportion of individuals who reported experiencing CNP daily, weekly, or monthly. Thus, by February 2025, half of the Peruvian adult population had experienced neck pain at least once a month during the prior six months.

**Table 4. Multinomial logistic regression assessing the association between electronic screen viewing and chronic neck pain intensity.**

| Chronic neck pain categories | OR | 95% Confidence interval | | p value |
|---|---|---|---|---|
| | | LL | UL | |
| **No, I have never had neck pain or have had it rarely** | Ref | | | |
| **Yes, I have had neck pain once a month on average** | | | | |
| Screen viewing | | | | |
| Rarely or never uses screens | 1.00 | | | |
| 1 to <4 hours/day | 1.43 | 0.76 | 2.70 | 0.266 |
| 4–8 hours/day | 2.56 | 1.30 | 5.04 | 0.007 |
| >8 hours/day | 3.02 | 1.39 | 6.56 | 0.005 |
| Age group (years) | | | | |
| 18 to <30 | 1.00 | | | |
| 30 to <60 | 1.22 | 0.81 | 1.82 | 0.345 |
| ≥60 | 1.26 | 0.71 | 2.26 | 0.429 |
| Sex | | | | |
| Male | 1.00 | | | |
| Female | 1.28 | 0.90 | 1.83 | 0.174 |
| Macroregion | | | | |
| Lima | 1.00 | | | |
| Northern | 1.08 | 0.69 | 1.69 | 0.724 |
| Central | 0.97 | 0.53 | 1.77 | 0.917 |
| Southern | 0.78 | 0.44 | 1.39 | 0.403 |
| Eastern | 0.64 | 0.32 | 1.26 | 0.193 |
| Constant | 0.21 | 0.10 | 0.42 | <0.001 |
| **Yes, I have had neck pain once a week on average** | | | | |
| Screen viewing | | | | |
| Rarely or never uses screens | 1.00 | | | |
| 1 to <4 hours/day | 1.88 | 0.90 | 3.96 | 0.095 |
| 4–8 hours/day | 3.44 | 1.53 | 7.71 | 0.003 |
| >8 hours/day | 3.87 | 1.63 | 9.15 | 0.002 |
| Age group (years) | | | | |
| 18 to <30 | 1.00 | | | |
| 30 to <60 | 1.09 | 0.69 | 1.72 | 0.703 |
| ≥60 | 1.17 | 0.61 | 2.27 | 0.634 |
| Sex | | | | |
| Male | 1.00 | | | |
| Female | 1.67 | 1.12 | 2.51 | 0.012 |
| Macroregion | | | | |
| Lima | 1.00 | | | |
| Northern | 1.52 | 0.94 | 2.45 | 0.090 |
| Central | 1.44 | 0.75 | 2.77 | 0.273 |
| Southern | 0.89 | 0.48 | 1.67 | 0.724 |
| Eastern | 0.80 | 0.38 | 1.70 | 0.567 |
| Constant | 0.09 | 0.04 | 0.22 | <0.001 |
| **Yes, I have had neck pain daily or almost daily** | | | | |
| Screen viewing | | | | |
| Rarely or never uses screens | 1.00 | | | |

*(Continued)*

**Table 4.** (Continued)

| Chronic neck pain categories | OR | 95% Confidence interval | | p value |
|---|---|---|---|---|
| | | LL | UL | |
| 1 to <4 hours/day | 1.47 | 0.80 | 2.72 | 0.216 |
| 4–8 hours/day | 1.85 | 0.89 | 3.81 | 0.097 |
| >8 hours/day | 4.17 | 1.90 | 9.13 | <0.001 |
| Age group (years) | | | | |
| 18 to <30 | 1.00 | | | |
| 30 to <60 | 2.23 | 1.37 | 3.63 | <0.001 |
| ≥60 | 4.06 | 2.15 | 7.69 | <0.001 |
| Sex | | | | |
| Male | 1.00 | | | |
| Female | 2.14 | 1.43 | 3.21 | <0.001 |
| Macroregion | | | | |
| Lima | 1.00 | | | |
| Northern | 1.51 | 0.91 | 2.50 | 0.107 |
| Central | 1.41 | 0.72 | 2.78 | 0.314 |
| Southern | 0.90 | 0.49 | 1.65 | 0.732 |
| Eastern | 2.96 | 1.69 | 5.21 | <0.001 |
| Constant | 0.05 | 0.02 | 0.11 | <0.001 |

LL: lower limit, UL: upper limit. ORa: adjusted odds ratio. Nagelkerke pseudo-R-square = 0.104, Variable inflation factor (VIF) for screen use = 1.14, VIF for age group = 1.13, VIF for macroregion = 1.02, VIF for sex = 1.00.

Our findings are consistent with those reported in studies based on data from GBD study. However, we must emphasize that the GBP defines neck pain as pain located from the occiput to the first thoracic vertebra that lasts for at least one day. On the basis of these criteria, the number of prevalent cases of chronic pain is projected to increase by 32.5% from 2020 to 2050 [2], whereas the age-standardized incidence will remain relatively stable [29]. In this comparison, we note that the definition used by GBD does not allow for the assessment of the chronicity of neck pain.

We explored the trend of CNP prevalence before and after the start of the social restrictions due to the pandemic and two years after. In a previous study in Peru that used the same methodology employed in this research, the estimated prevalence of daily or almost daily and at least weekly CNP in November 2022 was 2.43 and 2.25 times greater than the prepandemic prevalence, respectively [11]. However, in the period following the end of social restrictions and up to two years later, the increase was less pronounced and non-significant. In the abovementioned study [11], the prepandemic prevalence could have been affected by recall bias. However, the occurrence of an unusual historical event—such as the onset of mandatory social restrictions due to a pandemic—could have made it easier for respondents to recall what happened from that point backward with greater precision [30]. Another explanation is that a sedentary lifestyle and increased work at home contributed to a marked increase in the prevalence of CNP at different frequencies. The return to normality could have slowed the rate of increase in the prevalence of CNP, and the slight increase identified could be explained by other causes. This study specifically examined the potential effect of screen viewing on electronic devices.

We found that viewing electronic device screens for > 8 hours/day was strongly associated with experiencing CNP at all frequencies —daily, once a week on average, and once a month on average–. Moreover, screen viewing from even 4–8 hours/day increased the probability of CNP once a week on average and once a month on average, although not daily. These findings are consistent with those reported in Peru one month after the end of COVID-19-related social restrictions.

In that study, screen use > 8 hours/day was associated with an increased risk of experiencing CNP at more frequent levels compared with no screen use. This association was independent of age and sex [11]. Similar findings have been consistently reported in studies conducted among students and various occupational groups included in a meta-analysis. That study found that compared with individuals who did not view screens, those who used cell phone had 82% higher odds of experiencing CNP, whereas looking at computer screens was associated with a 23% increase in this risk. Furthermore, consistent with our results, longer screen exposure times were associated with an increased risk of neck pain [31].

The exposure analysed also changed during the study period, we identified changes in screen use patterns between November 2022 and February 2025. In general, the proportion of Peruvian adults who view electronic device screens has increased, particularly among those who use screens for 1 to less than 4 hours per day. In contrast, the proportion of individuals who spend more than 8 hours per day viewing screens has remained relatively stable at approximately 12%. The rise in electronic device screen use is not an isolated phenomenon in Peru; rather, it represents a global trend documented across various countries and population groups in the post–COVID-19 period, including adolescents [32], university-aged young adults [23], and the general population [33]. This increase is likely attributable to lifestyle changes and the growing reliance on technology for both work-related and recreational purposes. Such sustained exposure may help explain the persistence of CNP prevalence in Peru since the lifting of pandemic-related social restrictions. An alternative explanation could be population ageing; however, given the short two-year interval, this is unlikely to be a meaningful contributing factor.

Another important finding was the higher probability of CNP among women, specifically for weekly and daily CNP. This result is consistent with the findings at the population level in 2022 [11] and from a population study in Spain in 2017 [34]. In addition, according to the data from the GBD from 1990 to 2021, women consistently experienced a greater burden of neck pain in all age groups [3]. These epidemiological data are supported biologically by the results of animal experiments demonstrating sexual dimorphism in pain physiology. The underlying mechanisms involve the effects of oestrogen and its receptors; however, the multiple activated signalling pathways are complex, capable of either alleviating or exacerbating the perception of pain [35]. Another mechanism involved is related to inflammation, in which oestrogens have been proposed to contribute to inflammatory pain [35] through the activity of Toll-like receptor 4 (TRL4) and its signalling cascades [36]. In addition to biological factors, social factors and lifestyle play important roles, especially because of the greater sedentary lifestyle and greater burden of depression observed in women. Evidence indicates that interventions based on physical activity and exercise may be beneficial in reducing chronic pain [37]. Likewise, it has also been suggested that CNP may be associated with depression, mediated by factors such as reduced physical activity and poor sleep quality [38].

Our results also revealed a clear association between age and CNP. In the multivariable analysis, compared with individuals aged 18 to <30 years, those aged 30 to <60 years had twice the probability of experiencing daily CNP, while those aged ≥60 years had a fourfold higher probability. Evidence from the GBD study revealed that the prevalence of neck pain increases progressively up to the age of 70 years and subsequently decreases [2]. Although we were unable to corroborate the decrease in prevalence among those aged 70 and over, stratified analysis revealed a positive dose–response effect between screen use and CNP among those aged 18 to < 30 years but not among those aged ≥60 years. These age-specific differences may reflect confounding by age. Compared with individuals aged 18 to <30 years, those aged ≥ 60 years were more likely to report not viewing screens and less likely to spend ≥4 hours per day of screen viewing. This pattern could partially account for the lower prevalence of CNP observed among individuals aged ≥70 years in the GBD estimates. Conversely, the daily activities and lifestyle behaviors of younger adults may also contribute to this finding.

The 18- to 30-year-old population performs various occupations that increase the risk of musculoskeletal disorders, including neck pain. Additionally, the risk of neck pain due to a sedentary lifestyle generated by work activities is high among economically active adults [31]. A duration of sedentary activities of more than 4 hours per day increases the risk of neck pain by 60%, and the risk of experiencing neck pain increases with more hours spent viewing screens [31]. By

2024, 97% of the population of Peru aged 19–24 years were internet users, followed by those aged 25–40 years (92%), and daily internet access is usually achieved with a cell phone [39]. In our study, we found that half of the people aged 18 to < 30 years viewed electronic device screens for 4 to more hours per day, and one in five reported viewed screens ≥8 hours. A magnetic resonance imaging study demonstrated that excessive mobile phone use among adults aged 18–44 was associated with a higher likelihood of cervical disc degeneration, independent of sedentary occupational activity and age [40]. Although the most intensive screen use has declined—particularly among Peruvians aged 18 to <30 years—prolonged screen exposure remains a consistent risk factor for CNP in young adults.

Another important finding was the relatively high prevalence of CNP in the eastern region. In Peru, this region includes departments of the Amazon, where, owing to economic and climatic factors, motorcycles are the primary means of transportation. By 2023, national data in Peru indicated that up to 52% of the households in these departments owned a motorcycle. Worldwide, neck pain is a prevalent condition among drivers of these vehicles. For example, among 160 apparently healthy male motorcycle drivers between the ages of 20 and 29 in India, 68.7% reported neck pain [41]. Another study involving 473 adult motorcycle drivers in Indonesia reported that 67% experienced neck pain [42]. Finally, in Thailand, among the 448 motorcycle drivers dedicated full-time to meal delivery, 58.5% reported neck pain in the prior 12 months and that the number of hours driving increased the risk of neck pain [43]. These findings reflect an occupational risk for CNP and a potential interaction with electronic screen use in its development. Epidemiological studies are needed to confirm how riding motorcycles increases the risk of musculoskeletal disorders in residents of the Peruvian Amazon.

This study has several limitations. First, both the exposure and outcome variables were self-reported. CNP was not diagnosed through a clinical examination performed by healthcare professionals; however, the question used has been applied in previous research, enabling valid comparisons. Notably, many epidemiological studies on neck pain rely on self-reported measures that capture specific dimensions of the condition. In our study, we were able to characterize both the chronicity of the pain and its frequency pattern. Nonetheless, the self-reported assessment of CNP is particularly susceptible to recall bias, as respondents were asked to report symptoms occurring over the previous six months. Furthermore, precision in exposure measurement could be enhanced by differentiating the context of use (occupational versus recreational) and the type of device employed (tablets, smartphones, or computers). However, for the purposes of our study, such distinctions were not necessary, as we focused on evaluating the association between chronic neck pain and the static, sustained posture of the head and neck, irrespective of the usage context.

Another limitation, inherent to the cross-sectional design, is the inability to establish the temporal sequence between electronic device use and the development of CNP. Given the nature of these variables, reverse causality cannot be ruled out. Nevertheless, despite a reduction in screen exposure compared with 2022, it is reasonable to assume that screen viewing is a stable behavioral habit over time—likely preceding the onset of neck pain and contributing to its chronicity. Prospective studies would allow for direct estimation of the population-level risk of chronic neck pain, as well as risks within subgroups exposed to screen viewing.

Finally, as in the previous study, we were unable to adjust for several potential confounders, such as occupation, educational level, time spent in sedentary activities, physical activity, and history of trauma. We were unable to measure these variables due to logistical constraints inherent to the population survey. To enhance the internal validity of studies evaluating the association between screen viewing and CNP, we recommend adjusting for these factors and incorporating items from validated instruments to assess physical activity and occupational type.

Despite these limitations, the study has several strengths. The sampling design provided nationally representative estimates for Peru, encompassing both urban and rural areas, and enabled valid comparisons with our previous study, which assessed the prevalence of CNP following the lifting of pandemic-related social restrictions. This study represents a two-year follow-up conducted under conditions of normalcy, capturing subtle changes in both CNP prevalence and screen viewing patterns. Furthermore, CNP was assessed with the same methodology used in the previous study; this strategy

allowed us to measure pain specifically in the cervical region for a period of six months and to determine chronicity using response categories that reflect the pattern of occurrence over time.

## Conclusions

Two years after the lifting of COVID-19–related social restrictions, daily or almost daily chronic neck pain (CNP) affected 15.6% of the Peruvian population. This prevalence represents an increase of less than one percentage point compared with the estimate obtained immediately after the end of those restrictions—an increment that may fall within the margin of statistical error. Therefore, our findings confirm that CNP remains a prevalent public health problem in Peru, with no evidence of a reduction in its frequency. Although the proportion of individuals reporting screen use decreased slightly —particularly at the highest levels of exposure— the patterns observed in this study meet several criteria to be considered contributing factors to the occurrence of CNP. The association between screen use and CNP was most evident among adults aged 18 to <30 years, whereas other factors may play a more important role among those aged 60 years or older. Given that prolonged screen viewing is a modifiable risk factor, reducing excessive exposure—both at work and during leisure time—should be prioritized within comprehensive prevention strategies, as it may help lower the prevalence of CNP in the population. At the individual level, physicians should assess patients' daily screen-use intensity when evaluating persistent or recurrent neck pain and recommend appropriate postural hygiene measures, including avoiding excessive and sustained cervical flexion and minimizing sedentary behaviour characterized by prolonged static neck postures.

## Supporting information

**S1 Appendix. Raw database.**
(XLS)

## Acknowledgments

To Ipsos Peru for providing technical details of the survey administration service at the national level.

## Author contributions

**Conceptualization:** Tomas Nakazato, Franco Romani, César Gutiérrez.

**Data curation:** Franco Romani.

**Formal analysis:** Franco Romani.

**Funding acquisition:** Tomas Nakazato.

**Investigation:** Tomas Nakazato, Franco Romani, César Gutiérrez.

**Methodology:** Tomas Nakazato, Franco Romani, César Gutiérrez.

**Project administration:** Tomas Nakazato.

**Software:** Franco Romani.

**Supervision:** Tomas Nakazato, César Gutiérrez.

**Validation:** Tomas Nakazato, Franco Romani, César Gutiérrez.

**Visualization:** Franco Romani, Cesar Gutierrez.

**Writing – original draft:** Franco Romani.

**Writing – review & editing:** Tomás Nakazato, Franco Romani, Cesar Gutierrez.

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
