## [Decision Letter · Decision Letter 0]

5 Dec 2025

Dear Dr. Romani,

Thank you for submitting your manuscript to PLOS ONE. After careful consideration, we feel that it has merit but does not fully meet PLOS ONE’s publication criteria as it currently stands. Therefore, we invite you to submit a revised version of the manuscript that addresses the points raised during the review process.

We look forward to receiving your revised manuscript.

Kind regards,

Ravi Shankar Reddy, Ph.D

Academic Editor

PLOS One

**Journal Requirements:**

2. In the ethics statement in the Methods, you have specified that verbal consent was obtained. Please provide additional details regarding how this consent was documented and witnessed, and state whether this was approved by the IRB.

“This research was funding by Universidad de Piura (proyect code: PI2505).”

4. We are unable to open your Supporting Information file “Database.dta.” Please kindly revise as necessary and re-upload.

**Additional Editor Comments:**

The manuscript would benefit from addressing several key methodological and reporting issues. First, the authors should clarify the potential for recall bias in the comparison of pre-pandemic and current chronic neck pain (CNP) prevalence, as these are based on retrospective self-reports. Given the cross-sectional design, the discussion should avoid causal language when describing the association between screen time and CNP and instead emphasize the limitations in establishing temporality. The regression models did not control for several relevant confounders such as occupation, educational level, physical activity, and ergonomic factors, which could influence both exposure and outcome; this omission should be acknowledged explicitly. The definition of screen time should be more precise—specifying whether it includes both work and leisure use—to aid in interpretation. The rejection of the proportional odds assumption and the choice of multinomial logistic regression should be accompanied by the Wald test result or p-value for transparency. The outcome variable merges neck and shoulder pain into a single question, which may conflate clinically distinct conditions; the authors should justify this choice or consider separating them in future work. Tables would benefit from clearer presentation, including consistent formatting and complete confidence intervals. The manuscript should also state whether the survey questions were drawn from validated instruments or pilot tested. In the ethics section, the use of verbal consent should be confirmed as compliant with journal standards, including whether it was documented. Lastly, the increase in screen time should be contextualized by discussing whether it reflects behavioral shifts, infrastructure changes, or broader access to digital technology in Peru.

Reviewers' comments:

Reviewer's Responses to Questions

**Comments to the Author**

1. Is the manuscript technically sound, and do the data support the conclusions?

Reviewer #1: Yes

Reviewer #2: Yes

2. Has the statistical analysis been performed appropriately and rigorously?

Reviewer #1: Yes

Reviewer #2: N/A

3. Have the authors made all data underlying the findings in their manuscript fully available?

Reviewer #1: Yes

Reviewer #2: Yes

4. Is the manuscript presented in an intelligible fashion and written in standard English?

Reviewer #1: Yes

Reviewer #2: Yes

Reviewer #1: The manuscript provides timely and relevant evidence on the prevalence of chronic neck pain in Peru two years after the end of COVID-19–related social restrictions, and it offers clear public health implications. The study benefits from a nationally representative sample and a robust sampling strategy, which adds credibility to the findings. However, several methodological limitations warrant attention. Both the exposure (screen time) and the outcome (chronic neck pain) rely entirely on self-reported data, which may introduce recall bias and misclassification. Although the authors acknowledge this issue, a more detailed consideration of its potential impact on effect estimates would strengthen the manuscript. Additionally, important confounders such as occupation, ergonomic conditions, physical activity, and mental health variables were not measured or adjusted for, which limits the causal interpretability of the reported associations. The discussion effectively contextualizes the findings within global literature, though certain sections are overly extended and could be streamlined without losing analytical depth.

Reviewer #2: 1) Title and Abstract:

a) The title reflects the content of the study well, but the phrase "Two years after the COVID-19 pandemic" could be clarified (for example, the term "Post-Pandemic Period" could be more academic).

b) The abstract uses the term "multinomial logistic regression," but only the odds ratio (OR) is given in the results section. The reference category used (for example, those with no pain at all) should be briefly stated in the abstract.

c) For the phrase "slight but non-significant increase" in the results section, the p-value must be included in parentheses.

d) The conclusion in the conclusion focuses solely on prevalence. A sentence should be added explaining the public health implications of the finding regarding screen time.

2) Method

a) Screen Time: Screen time is categorized (<1, 1-4, 4-8, >8 hours). The basis for determining these cut-off points (prior literature or quartiles of the distribution) should be explained. It should be discussed whether they were analyzed as a continuous variable.

b) Confounders: Age, gender, region, and socioeconomic status (SEL) were included in the model. However, the reason why important variables that could directly affect neck pain, such as occupation or physical activity level, were not collected or included in the analysis should be explained in the "Limitations" section or in the Method.

3) Statistical Analysis:

a) The statistical test used to compare the 2022 and 2025 data (for example, two proportion difference tests or a regression model that includes the year variable) should be clarified.

b) Whether goodness-of-fit tests were performed in the regression model should be stated.

4) Discussion

a) (Limitations):

The self-reported nature of the data is the biggest limitation. The possibility of this creating recall bias should be emphasized more strongly.

5) Conclusion

Applicability: The results should not only assess the situation but also provide concrete recommendations for policymakers or healthcare professionals (e.g., implementing screen breaks in workplaces, etc.).

Future Studies: The need for incidence studies (cohorts), not just prevalence studies, should be emphasized.

.

Reviewer #1: **Yes:** Shahid AfridiShahid AfridiShahid AfridiShahid Afridi

Reviewer #2: **Yes:** Esedullah AKARASEsedullah AKARASEsedullah AKARASEsedullah AKARAS

---

## [Author Response · Author response to Decision Letter 1]

15 Dec 2025

Rebuttal letter

Dear Editor,

We would like to express our gratitude for the comments provided on our manuscript, which have helped improve its overall quality. Below, we address each of the reviewers’ observations.

Journal Requirements

We have reviewed the provided link and verified that the indicated requirements have been met.

2. In the ethics statement in the Methods, you have specified that verbal consent was obtained. Please provide additional details regarding how this consent was documented and witnessed, and state whether this was approved by the IRB.

We have included the requested information. The text has been added to the Ethics section of the Methods (lines 236-241).

The study protocol and the use of verbal informed consent were approved by the Institutional Research Ethics Committee of the University of Piura (file N ° CIEI.0039.01). Interviewers obtained verbal informed consent from all participants before data collection. Participants indicated their agreement to proceed with the interview by responding ‘yes’ or ‘no,’ and their responses were recorded in the iField application on a portable device.

“This research was funding by Universidad de Piura (proyect code: PI2505).”

We have incorporated this information into the requested documents.

4. We are unable to open your Supporting Information file “Database.dta.” Please kindly revise as necessary and re-upload.

We would like to offer our apologies; we are resending the dataset.

We appreciate the guidance. As a result of the review, only one additional reference needed to be added.

Editor comments

1. The manuscript would benefit from addressing several key methodological and reporting issues. First, the authors should clarify the potential for recall bias in the comparison of pre-pandemic and current chronic neck pain (CNP) prevalence, as these are based on retrospective self-reports.

We appreciate the observation. The reviewers noted this issue, and we have incorporated the limitation related to recall bias in the limitations section (lines 443-449)

First, both the exposure and outcome variables were self-reported. CNP was not diagnosed through a clinical examination performed by healthcare professionals; however, the question used has been applied in previous research, enabling valid comparisons. Notably, many epidemiological studies on neck pain rely on self-reported measures that capture specific dimensions of the condition. In our study, we were able to characterize both the chronicity of the pain and its frequency pattern. Nonetheless, the self-reported assessment of CNP is particularly susceptible to recall bias, as respondents were asked to report symptoms occurring over the previous six months.

2. Given the cross-sectional design, the discussion should avoid causal language when describing the association between screen time and CNP and instead emphasize the limitations in establishing temporality.

We reviewed the discussion section to avoid causal language between the variables of interest. In addition, this issue was acknowledged in the limitations (lines 454-459).

3. The regression models did not control for several relevant confounders such as occupation, educational level, physical activity, and ergonomic factors, which could influence both exposure and outcome; this omission should be acknowledged explicitly.

We appreciate the observation; we have explicitly acknowledged this limitation (lines 461–465), and it has been addressed in the discussion:

Finally, as in the previous study, we were unable to adjust for several potential confounders, such as occupation, educational level, time spent in sedentary activities, physical activity, and history of trauma. We were unable to measure these variables due to logistical constraints inherent to the population survey. To enhance the internal validity of studies evaluating the association between screen viewing and CNP, we recommend adjusting for these factors and incorporating items from validated instruments to assess physical activity and occupational type.

4. The definition of screen time should be more precise—specifying whether it includes both work and leisure use—to aid in interpretation.

We appreciate the observation. The operational definition described in the Methods section includes the question, which specifies the reason for screen use—both for work and recreational purposes. We recognize that not conducting a more precise measurement of the motivation for screen use may be a limitation. However, our question was designed with the primary objective of determining the association of neck pain with prolonged use of Internet-connected devices, all of which are recognized factors contributing to cervical pain. Accordingly, the survey did not differentiate whether screen time was attributable to occupational or recreational activities, as the aim of the present study was to evaluate the potential association between neck pain and sustained static head and neck posture, irrespective of the purpose of screen exposure.

In the limitations section (lines 4469-452), we have added that it would be advisable to improve the precision of exposure measurement by distinguishing the context that motivates use (work vs recreational) and the type of device used (tablets, mobile phones, or computers).

5. The rejection of the proportional odds assumption and the choice of multinomial logistic regression should be accompanied by the Wald test result or p-value for transparency.

We appreciate the comment and have incorporated this result as suggested by one of the reviewers (line 227).

6. The outcome variable merges neck and shoulder pain into a single question, which may conflate clinically distinct conditions; the authors should justify this choice or consider separating them in future work.

We appreciate the observation. The main variable was assessed using the question: “Within the last six months, have you had pain in the neck, in the nape, and/or above your shoulders?” We corrected the translation of the original Spanish question to more accurately reflect its intended meaning.

This question identifies the occurrence of neck pain according to the criteria of the Neck Pain Task Force. It specifically targets pain in the upper shoulder region rather than the shoulder joint itself. The wording was designed to include the area above the clavicle, the acromion, and the scapular spine, with the aim of detecting involvement of the upper trapezius and levator scapulae muscles, which are common sources of neck pain. It intentionally excludes references to pain in the infraspinatus or deltoid muscles, which are more closely associated with shoulder joint pathology.

7. Tables would benefit from clearer presentation, including consistent formatting and complete confidence intervals.

We appreciate the comment. We have added the confidence intervals in Table 2. Additionally, we have omitted the presentation of the standard error. The tables now report the estimate, the confidence interval, and the p-value (when applicable).

8. The manuscript should also state whether the survey questions were drawn from validated instruments or pilot tested.

The questions used to measure chronic neck pain and the exposure were single-item measures and were not subjected to formal psychometric validation. However, both items were previously used in an epidemiological study conducted in 2016 and again in 2022, providing indirect evidence of their adequacy and usability at the population level.

We have incorporated this explanation in the first paragraph of the ‘Instrument and Variables’ subsection (lines 137–140)

9. In the ethics section, the use of verbal consent should be confirmed as compliant with journal standards, including whether it was documented.

We have included the requested information. The text has been added to the Ethics section of the Methods (lines 236-241).

The study protocol and the use of verbal informed consent were approved by the Institutional Research Ethics Committee of the University of Piura (file N ° CIEI.0039.01). Interviewers obtained verbal informed consent from all participants before data collection. Participants indicated their agreement to proceed with the interview by responding ‘yes’ or ‘no,’ and their responses were recorded in the iField application on a portable device.

10. Lastly, the increase in screen time should be contextualized by discussing whether it reflects behavioral shifts, infrastructure changes, or broader access to digital technology in Peru.

We have briefly added that the slight increase in screen-viewing intensity is primarily attributable to lifestyle changes (lines 387–388). Between 2022 and 2025, access to internet connectivity and digital technology in Peru has not undergone substantial changes.

Reviewer #1

1. The manuscript provides timely and relevant evidence on the prevalence of chronic neck pain in Peru two years after the end of COVID-19–related social restrictions, and it offers clear public health implications. The study benefits from a nationally representative sample and a robust sampling strategy, which adds credibility to the findings. However, several methodological limitations warrant attention. Both the exposure (screen time) and the outcome (chronic neck pain) rely entirely on self-reported data, which may introduce recall bias and misclassification. Although the authors acknowledge this issue, a more detailed consideration of its potential impact on effect estimates would strengthen the manuscript.

We appreciate the observation, which was also raised by Reviewer 2. We have expanded on these study limitations in the Discussion section (lines 443–452). Additionally, we have clarified in the Methods section that this measurement strategy was employed in previous studies, allowing for valid comparisons with those earlier findings.

2. Additionally, important confounders such as occupation, ergonomic conditions, physical activity, and mental health variables were not measured or adjusted for, which limits the causal interpretability of the reported associations.

We appreciate the comments. This aspect has also been further elaborated in the study’s limitations section (lines 461–465):

Finally, as in the previous study, we were unable to adjust for several potential confounders, such as occupation, educational level, time spent in sedentary activities, physical activity, and history of trauma. We were unable to measure these variables due to logistical constraints inherent to the population survey. To enhance the internal validity of studies evaluating the association between screen viewing and CNP, we recommend adjusting for these factors and incorporating items from validated instruments to assess physical activity and occupational type.

3. The discussion effectively contextualizes the findings within global literature, though certain sections are overly extended and could be streamlined without losing analytical depth.

We have reviewed the text to reduce the length of some paragraphs in the discussion.

Reviewer #2

Title and Abstract:

1. The title reflects the content of the study well, but the phrase "Two years after the COVID-19 pandemic" could be clarified (for example, the term "Post-Pandemic Period" could be more academic).

We appreciate the observation; we have made the correction as suggested by the reviewer.

2. The abstract uses the term "multinomial logistic regression," but only the odds ratio (OR) is given in the results section. The reference category used (for example, those with no pain at all) should be briefly stated in the abstract.

We appreciate the observation; we have made the correction as suggested by the reviewer.

3. For the phrase "slight but non-significant increase" in the results section, the p-value must be included in parentheses.

We have added the point estimates and confidence intervals for both observation points. In addition, we have included the p-values corresponding to the comparison of proportions using the chi-square test with second-order Rao–Scott correction, taking into account the survey sampling weights.

4. The conclusion in the conclusion focuses solely on prevalence. A sentence should be added explaining the public health implications of the finding regarding screen time.

We appreciate the observation; we have made the correction as suggested by the reviewer (lines 484-489):

Given that prolonged screen viewing is a modifiable risk factor, reducing excessive exposure—both at work and during leisure time—should be prioritized within comprehensive prevention strategies, as it may help lower the prevalence of CNP in the population. At the individual level, physicians should assess patients’ daily screen-use intensity when evaluating persistent or recurrent neck pain and recommend appropriate postural hygiene measures, including avoiding excessive and sustained cervical flexion and minimizing sedentary behaviour characterized by prolonged static neck postures.

5. Screen Time: Screen time is categorized (<1, 1-4, 4-8, >8 hours). The basis for determining these cut-off points (prior literature or quartiles of the distribution) should be explained. It should be discussed whether they were analyzed as a continuous variable.

The exposure variable was assessed directly as a categorical measure. Cutoff points were selected to allow comparability with a prior study conducted in the same population. Our previous national survey research used four categories for all questions. To maintain consistency with this structure, we operationalized exposure using the following categories: <1 hour, 1–4 hours, 4–8 hours, and >8 hours per day, with the latter representing prolonged exposure. This approach was chosen because epidemiological studies in the literature use heterogeneous thresholds that vary by age group, duration of exposure relative to the outcome, and the context motivating screen use. Given our objective—to determine whether prolonged screen viewing is associated with neck pain—we defined exposure exceeding 8 hours per day as prolonged and therefore as a potential risk factor. This threshold was informed by recommendations limiting a standard workweek to 48 hours and previous studies (lines 156- 162).

We acknowledge that collecting the data as a continuous variable would have allowed greater analytical flexibility; however, expressing exposure as hours per day would not adequately reflect weekly or monthly variability, which could make reporting more difficult for participants.

6. Confounders: Age, gender, region, and socioeconomic status (SEL) were included in the model. However, the reason why important variables that could directly affect neck pain, such as occupation or physical activity level, were not collected or included in the analysis should be explained in the "Limitations" section or in the Method.

We appreciate the observation. This methodological aspect represents a limitation of the study. In the current version of our manuscript, we acknowledge this limitation (Lines 461–465). We were unable to inco

---

## [Decision Letter · Decision Letter 1]

18 Feb 2026

Screen time and chronic neck pain in Peru: A comparative population-based cross-sectional study in the COVID-19 post-pandemic period

PONE-D-25-60168R1

Dear Dr. Romani,

We’re pleased to inform you that your manuscript has been judged scientifically suitable for publication and will be formally accepted for publication once it meets all outstanding technical requirements.

Kind regards,

Armaan Jamal

Guest Editor

PLOS One

Reviewers' comments:

Reviewer's Responses to Questions

**Comments to the Author**

Reviewer #2: All comments have been addressed

2. Is the manuscript technically sound, and do the data support the conclusions?

Reviewer #2: Yes

3. Has the statistical analysis been performed appropriately and rigorously?

Reviewer #2: Yes

4. Have the authors made all data underlying the findings in their manuscript fully available?

Reviewer #2: Yes

5. Is the manuscript presented in an intelligible fashion and written in standard English?

Reviewer #2: Yes

Reviewer #2: (No Response)

.

Reviewer #2: **Yes:** Esedullah AKARASEsedullah AKARASEsedullah AKARASEsedullah AKARAS
